# Semantic Self-Consistency: Enhancing Language Model Reasoning via Semantic Weighting

**Tim Knappe**[*]   **Ryan Li**   **Ayush Chauhan**   **Kaylee Chhua**   **Kevin Zhu**[†]   **Sean O'Brien**[†]
Algoverse AI Research
`cs.timknappe@gmail.com`, `kevin@algoverse.us`

## Abstract

While large language models (LLMs) have rapidly improved their performance on a broad number of tasks, they still often fall short on reasoning tasks. As LLMs become more integrated in diverse real-world tasks, advancing their reasoning capabilities is crucial to their effectiveness in nuanced, complex problems. Wang et al. [33]'s *self-consistency* framework reveals that sampling multiple rationales before taking a majority vote reliably improves model performance across various closed-answer reasoning tasks. Standard methods based on this framework aggregate the final decisions of these rationales but fail to utilize the semantic information detailed in the step-by-step reasoning paths. Our work introduces *semantic self-consistency*, enhancing this approach by incorporating and analyzing both the reasoning paths of these rationales in addition to their final decisions before taking a majority vote. These methods not only improve the reliability of reasoning paths but also cause more robust performance on complex reasoning tasks.

## 1   Introduction

In recent years, the development of large language models has witnessed remarkable strides, with significant advancements in their accuracy and expressive capabilities [3, 28, 24, 4]. Despite these achievements, models still perform suboptimally in domains such as mathematics, commonsense, and complex algorithmic reasoning [10]. Various methods such as *chain-of-thought* prompting have been developed to further increase reasoning capabilities and was further enhanced by the introduction of self-consistency, which demonstrate that baselines can be pushed forward by sampling and ensembling multiple model responses with chain-of-thought to improve prediction quality [34, 23].

We build on the framework of self-consistency, proposing two techniques that add a separate semantic weighting step to rerank results based on their reasoning paths. To achieve this, we use semantic vector embeddings in combination with self-consistency to group consistent model outputs, aiding in the identification of similar responses to estimate the most likely output. Additionally, we introduce a semantic filtering mechanism that discards degenerate or hallucinated outputs, which can be utilized for analyzing smaller sample sizes. Overall, we demonstrate that self-consistency with semantic marginalization not only improves accuracy across a range of benchmarks but also serves as a filtering mechanism. By introducing these methods, we aim to provide a framework for improving performance and analyzing the semantic usage of model outputs in reasoning.

---

[*]Lead Author
[†]Senior Author

38th Conference on Neural Information Processing Systems (NeurIPS 2024).

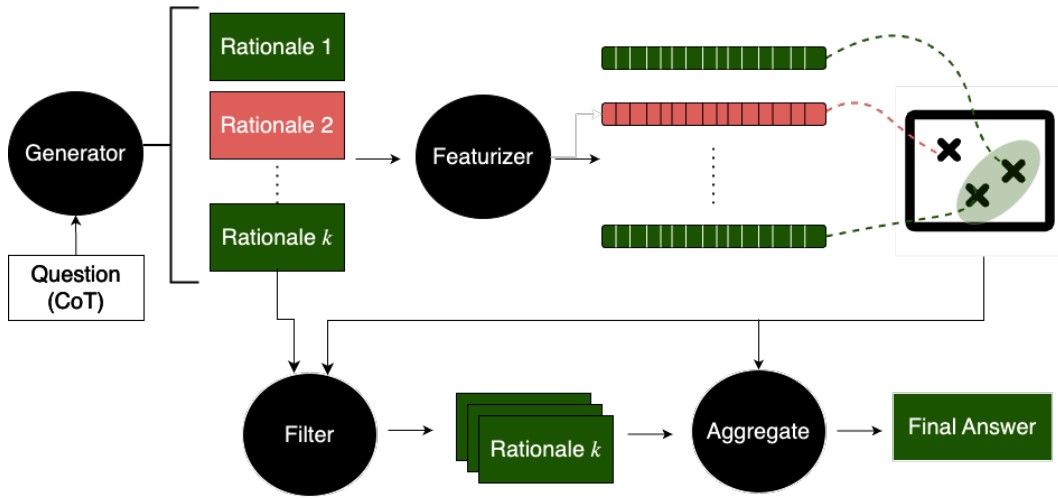

Figure 1: Whereas baseline self-consistency comprises three steps: (1) Prompt a model with chain-of-thought, (2) generate $n$ sampled sequences, and (3) choose results based on the most occurring final output, our proposed method, shown above, decides based on the semantic consistency of the employed reasoning path. Our assumption is that language models often apply the correct reasoning but lack the ability to conclude to the correct result.

## 2  Datasets

We evaluate the models on arithmetic and commonsense reasoning using three datasets: **AQuA-RAT**, **SVAMP**, and **StrategyQA**. **AQuA-RAT** assesses models' ability to solve arithmetic problems involving basic calculations, numerical relationships, and multi-step reasoning [17]. **SVAMP** challenges models with math problems focused on algebraic manipulations and symbolic reasoning [25]. **StrategyQA** tests models on answering complex, open-domain questions that require strategic thinking rather than simple factual knowledge [9]. For specific information please refer to Appendix L.

## 3  Language Models

Our models are categorized into two types: *generators*, which produce sequences such as text, code, or reasoning steps, and *featurizers*, which transform these outputs into numerical representations (vector embeddings) that summarize their meaning for analysis.

Detailed information on the configurations used for our models can be found in Appendix I.3. Additionally important hyperparameters for different methods are discussed in Appendix I We use chain-of-thought prompting for all of our experiments. The prompts can be found in Appendix K.

### 3.1  Generators

For our evaluation, we use several models with varying architectures and sizes. First, we utilize **GPT-3.5**, a closed-source large-scale transformer model developed by OpenAI [3]. Additionally, we evaluate both **Llama 2** (7B parameters) [32] and **Llama 3** (8B parameters) [8], which are open-weight models known for their strong performance on numerous benchmarks. We also include **Mistral 7B** (version 0.1), recognized for its robust capabilities across a variety of language processing tasks [13]. Lastly, we assess **GPT-4o mini**, a lower parameter variant of the GPT-4o architecture that balances computational efficiency with high performance across diverse language tasks.

### 3.2  Featurizers

All of our featurizers are based on the **BERT** (Bidirectional Encoder Representations from Transformers) model architecture [7], with various fine-tuned versions used to generate embedding vectors

tailored to specific datasets. **RoBERTa** is employed for the StrategyQA dataset, which requires reading comprehension and contextual reasoning, benefiting from RoBERTa's robustness in general language processing tasks [19]. Additionally, we use **SciBERT** for the AQuA-RAT and SVAMP datasets, which focus on mathematical reasoning, as its specialization in scientific texts makes it well-suited to handle the language present in these datasets [2].

# 4 Methodology

We analyze three main mechanisms for weighting and categorization (CPW, sequence comparison, and filtering of anomalous points) that follow a similar operational pattern outlined below:

1. *Generate candidate responses:* Given a query of few-shot examples, we generate $n$ samples based on chain-of-thought prompting [34].

2. *Embed reasoning paths:* We represent each generated rationale as a vector embedding using fine-tuned BERT models (e.g., SciBERT for mathematical reasoning tasks). Instead of focusing on individual sentences or tokens, we obtain a single vector representation for each entire reasoning path, capturing its overall semantic content.

3. *Semantic consistency or outlier removal:* We apply various algorithms to weight and aggregate the responses based on their featurized embedding vectors, enhancing decision-making by emphasizing semantically consistent reasoning paths or removing outliers.

## 4.1 Semantic consistency

### 4.1.1 Centroid Proximity Weighting

In a set of examples, general answers often display similar patterns, suggesting the application of embedding vectors to map responses into an $n$-dimensional space. To identify the most relevant features, we first compute the centroid of the embeddings, centroid $= \frac{1}{N} \sum_{i=1}^{N} \text{data\_embedding}[i]$. Then, we calculate the distance of each vector from the centroid, distances$[i] = \|\text{data\_embeddings}[i] - \text{centroid}\|$, and normalize these distances, normalized\_distances$[i] = \frac{\text{distances}[i]}{\sum_{j=1}^{N} \text{distances}[j]}$. We assign weights to the vectors inversely proportional to their normalized distances, weights$[i] = \frac{1}{\text{normalized\_distances}[i]}$. Finally, the total weight for each unique output is computed as sum\_weights$[u] = \sum_{i \in I(u)} \text{weights}[i]$, where outputs with the highest total weights are considered the most likely to be correct.

### 4.1.2 Semantic Consensus Weighting

To compare the weighting of embedding positions, we introduce another method and weigh responses relative to their respective sequences with cosine similarity, a measurement of how similar two vectors. We take $n_1, n_2, n_3, \ldots, n_i$ as distinct elements in our set $N$, where each $n$ corresponds to a featurized embedding vector. The cosine similarity between vectors $n_a$ and $n_b$ is given by cosine\_similarity$(n_a, n_b) = \frac{n_a \cdot n_b}{\|n_a\|_2 \|n_b\|_2}$, and for each $n_e$, we compute the cosine similarity with every $n_i$ in $N$ and aggregate the scores: $S_{n_e} = \sum_{n_i \in N} \text{cosine\_similarity}(n_e, n_i)$. This process is repeated for each $n_j$ in $N$, resulting in aggregated scores $S_{n_1}, S_{n_2}, S_{n_3}, \ldots, S_{n_i}$, and the scores are summed based on their answer decision, leading to the selection of the highest consensual response.

## 4.2 Outlier removal

To eliminate outliers, we filter responses based on proximity [18, 21, 5], isolating data points that significantly deviate and identifying flawed reasoning, degenerated outputs, or model hallucinations. We examine the following common methods: (1) **K-nearest neighbor**, using $\sqrt{\sum_{i=1}^{n}(x_i - y_i)^2}$; (2) **Isolation forest**, where $s(x, n) = 2^{-\frac{E(h(x))}{c(n)}}$; and (3) **Support vector machines**, defined by $\frac{1}{2}\omega^T\omega + C\sum_{i=1}^{n} \zeta_i$.

# 5 Results

## 5.1 Semantic consistency results

We compared Centroid Proximity Weighting (CPW) and Semantic Consensus Weighting (SCW) with the self-consistency baseline across datasets. As shown in Table 1, SCW generally outperformed CPW. For Llama 2 7B, SCW boosted accuracy on StrategyQA by **13.53 %**, while CPW improved it by **6.11 %**. GPT 3.5 also saw a **7.89 %** gain with SCW, but CPW caused a **1.6%** drop. GPT-4o mini underperformed with CPW across all datasets. Cosine similarity improved most models, except Mistral 7B on StrategyQA and Llama 3 8B on SVAMP, while CPW underperformed in six out of fifteen model-dataset pairs.

| Dataset | Method/Metric | Llama 2 7B | Mistral 7B | GPT 3.5 | Llama 3 8B | GPT-4o mini |
|---|---|---|---|---|---|---|
| AQuA-RAT | Top prob sample | 21.65 | 24.34 | 53.63 | 43.02 | 79.22 |
| | SC baseline | 24.80 | 25.60 | 59.40 | 45.28 | 83.07 |
| | CPW | 24.60 (**-0.2**) | 29.00 (**+3.4**) | 68.00 (**+8.6**) | 46.06 (**+0.78**) | 82.68 (**-0.39**) |
| | SCW | 25.00 (**+0.2**) | 29.80 (**+4.2**) | 65.40 (**+6.0**) | 47.48 (**+2.2**) | 86.18 (**+3.11**) |
| SVAMP | Top prob sample | 31.90 | 65.18 | 77.42 | 70.55 | 85.62 |
| | SC baseline | 46.50 | 68.50 | 79.80 | 73.33 | 89.80 |
| | CPW | 47.40 (**+0.9**) | 69.80 (**+1.3**) | 81.00 (**+1.2**) | 74.67 (**+1.34**) | 89.60 (**-0.2**) |
| | SCW | 46.90 (**+0.4**) | 70.20 (**+1.7**) | 80.30 (**+0.5**) | 73.00 (**-0.33**) | 92.38 (**+2.98**) |
| StrategyQA | Top prob sample | 46.79 | 64.27 | 63.21 | 60.32 | 75.32 |
| | SC baseline | 48.91 | 67.98 | 66.81 | 63.32 | 79.18 |
| | CPW | 55.02 (**+6.11**) | 60.70 (**-7.28**) | 65.21 (**-1.6**) | 63.32 (**+0.0**) | 73.80 (**-5.38**) |
| | SCW | 62.44 (**+13.53**) | 65.35 (**-2.63**) | 74.70 (**+7.89**) | 71.47 (**+8.15**) | 79.68 (**+0.5**) |

Table 1: Accuracy comparison of CPW and cosine similarity on different datasets and models, with SciBERT embeddings for AQuA-RAT and SVAMP and RoBERTa encodings for StrategyQA.

CPW improved self-consistency by **3.14%** on AQuA-RAT and **0.97%** on SVAMP but decreased performance by **-1.63%** on StrategyQA, likely due to its limited reasoning paths. This effect was seen across self-consistency, where improvements were smaller compared to other datasets. A detailed discussion of these suboptimal cases is in Appendix D.

SCW showed that weighting sequences based on consistency reduces errors and improves accuracy, outperforming baseline self-consistency.

## 5.2 Outlier detection results

The results from our analysis of various outlier detection methods isolation forest, k-nearest neighbor, one-class support vector machines (SVM) demonstrate their effectiveness in refining the quality of model output. The observed increases in accuracy across these methods remain consistent towards reduced sample sizes as well, suggesting that the effectiveness of anomaly detection techniques are not solely dependent on sample size. Obtained results exhibited slight deviations between the different configurations. A review across different sets of configurations and parameters can be found under Appendix I.2.1 to I.2.3.

The found results highlight variability across datasets, with isolation forest and one-class SVM performing better on certain datasets.

# 6 Discussion

It is worth noting that our system uses embedding vectors to filter responses based on general reasoning accuracy, prioritizing broad similarity over subtle variations, as the benefit of choosing the numerical majority vote from self-consistency to yield correct answers still applies, especially in the limited rationale space. An additional analysis can be found in Appendix B.

Diverse responses are not necessarily undesirable and can lead to elevated results as shown in Appendix G.1. Against the natural feel, employed methods do not discriminate against diverse reasoning. Lowering the temperature will make multiple responses more diverse and, therefore, broaden the distribution. This will not affect performance when using CPW or outlier detection, since

| Dataset | Method | Llama 2 | Mistral | GPT 3.5 | Llama 3 | GPT4o mini |
|---|---|---|---|---|---|---|
| | | Best / Average | | Best / Average | | Best / Average |
| AQuA-RAT | SC baseline | 24.8 / 24.8 | 25.6 / 25.6 | 59.4 / 59.4 | 45.28 / 45.28 | 83.07 / 83.07 |
| | Isolation Forest | **28.45 / 26.04** | **26.61 / 25.97** | **65.27 / 63.73** | **72.25 / 68.59** | 70.86 / 69.78 |
| | K-nearest neighbors | 25.40 / 25.37 | 25.91 / 25.66 | **62.81 / 60.04** | **68.10 / 66.74** | 71.65 / 70.81 |
| | One-class SVM | **26.70** / 24.25 | **28.45 / 26.08** | 59.55 / 59.26 | **68.39 / 65.91** | 70.87 / 69.23 |
| SVAMP | SC baseline | 46.5 / 46.5 | 68.5 / 68.5 | 79.8 / 79.8 | 73.33 / 73.33 | 89.80 / 89.80 |
| | Isolation Forest | 45.94 / 45.60 | 68.84 / 68.34 | **84.65 / 84.28** | **84.44 / 81.75** | 84.44 / 81.76 |
| | K-nearest neighbors | 45.85 / 45.71 | 68.84 / 68.52 | **84.64 / 84.42** | **82.57 / 81.85** | 82.57 / 81.85 |
| | One-class SVM | 44.94 / 43.30 | 67.23 / 65.33 | **85.23 / 84.54** | **82.11 / 80.70** | 82.11 / 80.70 |
| StrategyQA | SC baseline | 48.91 / 48.91 | 67.98 / 67.98 | 66.81 / 66.81 | 63.32 / 63.32 | 79.18 / 79.18 |
| | Isolation Forest | 49.34 / 49.01 | 68.70 / 68.13 | **70.07 / 69.01** | **70.80 / 69.37** | 79.91 / 79.56 |
| | K-nearest neighbors | 49.49 / 49.09 | **69.00 / 68.61** | 68.65 / 68.57 | **69.43 / 69.10** | 80.64 / 80.28 |
| | One-class SVM | **49.85** / 48.98 | **69.43 / 68.81** | 68.73 / 68.27 | **70.45 / 69.23** | **81.02 / 80.65** |

Table 2: Outlier detection performance on SVAMP, AQuA-RAT, and StrategyQA. Performance increase over baseline of $n > 1\%$ featured in bold. Encoded based on SciBERT for mathematical reasoning and RoBERTa for commonsense.

outputs farther from the mean are not outliers but sensible parts of a wider distribution. Consequently, the weighting process will remain consistent, as all values will proportionally receive lower weights.

# 7 Conclusion

Our investigation into weighting and anomaly detection methods shows that cosine similarity outperforms CPW in improving model accuracy, particularly for models like Llama 2 7B and GPT 3.5 on datasets such as StrategyQA. CPW was effective for AQuA-RAT and SVAMP, leading to accuracy increases, but less so for StrategyQA. Our system prioritizes general reasoning accuracy using embedding vectors, with numerical majority voting from self-consistency remaining a key factor in achieving correct answers, especially within limited rationale spaces. Please note that the recommended methods should be employed with carefully tested hyperparameters, as their effectiveness may vary with subtle implementation nuances.

# 8 Related Work

Reasoning is an ubiquitous issue across many domains. [6]. One significant advancement in the area has been the development of the chain-of-thought prompting [34, 27] and self-consistency [33], which we extend for our Method. Self improvement of Language Models after generation is a well-known method for improving accuracy [12]. This concept has often been adapted by other weighting methods during pre-training to improve overall accuracy [31, 20], using different methods to shift the distribution [14].

# 9 Limitations

Our study proposes the application of semantic vector representations to group and weigh model outputs, which is designed to facilitate the identification of consensus responses [33]. Semantic vectors must capture variations in meaning and context, which is particularly hard in abstract reasoning tasks without a sufficient amount of context making prompting techniques to enhance the models output structure and size an important factor as visualized in Table 3. The process of clustering based on semantic vectors can be challenging due to the nuanced and abstract nature of reasoning processes. This limitation underscores the need for advanced featurization models and explicit choice of a fitting fine-tuned model [22]. Like showcased in Table 6, multiple models should be considered for semantic analysis, to ensure that the model outputs are grouped in a way that truly reflects their underlying meaning and relevance. Without these fitting featurizers, on fields with more subtle variations or on short sequences, the employed method might not be able to distinguish different sequences well enough to uphold a notable positive effect.

## 10 Reproducibility Statement

Our experiments include a variety of models with different sizes. GPT 3.5 as well as GPT-4o mini have API endpoints that are open for public use `https://openai.com/blog/openai-api`.

Mistral 7B is available for unrestricted use under the Apache 2.0 license, while its model architecture and setup are open source: `https://github.com/Mistralai/Mistral-src`.

Llama 2 7B and Llama 3 8B are models with restricted access, made available by Meta. One can gain access to them by requesting permission through the provided Meta license. `https://ai.meta.com/llama/`.

All of our BERT models are built upon the BERT-base model developed by google-research, which is accessible under the Apache 2.0 license, including MathBERT and SciBERT. RoBERTa can be used under the MIT license.

Our datasets as well as the configurations used for our language models are accessible throughout this paper and in the Appendix to aid the reproducibility of our experiments.

### 10.1 GPU usage

| approx. Hours | GPU | Model | Memory |
|---|---|---|---|
| 250 h | NVIDIA | T4 | 15GB |
| 50 h | NVIDIA | V100 | 16GB |
| 60 h | NVIDIA | A100 | 40GB |
| 100 h | NVIDIA | TPU v2 | 32GB |

## 11 Ethical Considerations & Risks

Language models may produce factually incorrect or biased outputs based on user prompts. The BERT-based featurizers, trained on English corpora, may yield inconsistent results in other languages. Mistral 7B, Llama 2 7B, and Llama 3 8B lack built-in content moderation, needing external safeguards against harmful content. While GPT-4o and GPT-4o mini have stronger moderation, biases may still emerge.

Further risks include that embedding and clustering methods may introduce subtle biases by emphasizing specific response types over others. Additionally variations in model temperature and sampling can add unintended randomness. Controlled sampling and inverse temperature weighting help but require careful tuning.

We recommend using monitoring tools and responsible model deployment, particularly in high-stakes applications.

## 12 Acknowledgements

We thank Celine Lee and Andy Chung for their helpful feedback on the methods and assistance in improving the clarity of our work. We also thank the anonymous reviewers of the conference for their insightful comments and suggestions, which helped improve the quality of this paper.

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

# A    Performance variation

Across different findings, we see a variation in performance with a general upward trend. As shown in Section 3 and discussed in Appendix B, sequence length seems to affect model performance positively. Smaller sequences tend to contain to be less similar in terms of informational density compared to all other sequences.

Moreover, GPT 3.5's and GPT-4o mini's instruction fine-tuning positively affects sequence length and output content, leading to longer and more contextual sentences. Additionally, there's a trend towards larger models, suggesting that increased parameter size may improve performance across tasks and the way information is packed across the exemplars.

# B    Effects of symbolic logic and embeddings

Subtle variations in reasoning or content, particularly in fields like mathematics, can lead to significant divergences in outcomes, suggesting a preference for symbolic logic to distinguish these differences precisely. This approach presupposes that correct reasoning across various contexts tends to follow similar operational patterns. By leveraging embedding vectors, the system isolates responses that deviate significantly in reasoning quality or factuality, rather than getting entangled in the minutiae of every possible variation. Thus, while embedding vectors may overlook some subtle differences, their use is justified by their effectiveness in broadly categorizing and filtering responses according to general reasoning accuracy.

Additionally, the inherently delivered effect of self-consistency implies that multiple exemplars, when exhibiting correct or similar reasoning, will eventually result in the majority of correct numerical answers, which will prove especially effective when the space of rationales is limited to these that are sufficiently supported by its reasoning path.

We observe a slight correlation between the average sequence length generated by our models and improvements in accuracy, emphasizing the role of exemplar selection in the reasoning process. Longer chains of thought can provide more context, but they are also more prone to outliers and inaccuracies. Similarly, shorter sequences often lack sufficient context to differentiate responses effectively.

Although sequence length scales with accuracy, we observe no correlation between accuracy and the averaged BLEU score. This suggests that improvements in text generation quality, as measured by BLEU, do not necessarily translate to better reasoning accuracy, underscoring the trade-off between context depth and noise in model predictions.

| Dataset | Model | Avg. Seq. Length | Avg. Accuracy Increase (%) | Avg. BLEU Score |
|---------|-------|------------------|----------------------------|-----------------|
| AQuA-RAT | GPT 3.5 | 102.40 | 7.30 | 0.342 |
| AQuA-RAT | Mistral | 53.24 | 3.80 | 0.031 |
| AQuA-RAT | Llama 2 | 49.58 | 0.00 | 0.045 |
| AQuA-RAT | Llama 3 | 56.21 | 1.49 | 0.185 |
| AQuA-RAT | GPT-4o mini | 83.65 | 1.36 | 0.358 |
| SVAMP | GPT 3.5 | 49.71 | 0.85 | 0.440 |
| SVAMP | Mistral | 52.92 | 1.50 | 0.152 |
| SVAMP | Llama 2 | 52.29 | 0.65 | 0.213 |
| SVAMP | Llama 3 | 83.45 | 0.505 | 0.300 |
| SVAMP | GPT-4o mini | 80.32 | 1.19 | 0.547 |
| StrategyQA | GPT 3.5 | 92.66 | 3.145 | 0.289 |
| StrategyQA | Mistral | 50.68 | -4.955 | 0.227 |
| StrategyQA | Llama 2 | 60.39 | 9.82 | 0.075 |
| StrategyQA | Llama 3 | 77.84 | 4.075 | 0.141 |
| StrategyQA | GPT-4o mini | 88.91 | -2.44 | 0.327 |

Table 3: Comparison of Sequence Length, Accuracy Increase, and BLEU Score across datasets and models

Larger sequences initially perform better as they leverage more context, but this benefit diminishes as the sequence length grows too large, resulting in the loss of relevant information. Shorter sequences,

in contrast, often fail to provide enough context for the model to make accurate distinctions between responses. BLEU scores reveal that while text generation quality improves moderately with longer sequences, it does not strongly correlate with accuracy improvements. This highlights the trade-off between providing enough context and minimizing noise in model predictions [1].

| Model | SVAMP | AQuA-RAT | SQA |
|---|---|---|---|
| Mistral | 1.01 | 2.05 | 1.50 |
| Llama 2 | 2.13 | 0.90 | 1.25 |
| GPT 3.5 | 0.33 | 0.57 | 0.70 |
| Llama 3 | 1.20 | 1.80 | 1.40 |
| GPT-4o mini | 0.80 | 1.00 | 0.95 |

Table 4: Accuracy deviation (%) across models and datasets.

## C  Embedding quality analysis

It is important to distinguish that the employed system focuses on identifying consensual responses and broader similarity in the representational space of embeddings, rather then subtle nuances. A clear analysis of our embeddings in connection to symbolic logic and subtle details can be found in Appendix B.

To test our embeddings and ensure that embeddings do not solely discriminate on numerical output, we randomly removed numerical outputs before generating embedding vectors. As visible in the results, performance remained stable and proves that even correctly reasoned but arithmetically incorrect responses can still be used in different methods to enhance overall output quality and mechanisms that make use of semantic evaluation.

Further analysis of both the embedding distribution as well as our dimensionality reduction can be found in Appendix O.

## D  Self-consistency failure scenarios

Although we observe an upward trend in performance, there are certain scenarios where the applied methods fail to deliver the desired results.

- **Overly similar generations**: Generations that provide overly similar reasoning will likely be categorized in a similar position in the embedding space, which will lead to our semantic methods, not being able to discern between elements.
- **Small subtleties in generations**: As described in Section B & Section 6 small subtleties aren't captured directly by our model, making it less capable in tasks like Symbolic Reasoning or state tracking.

## E  Comparison to related Methods

### E.1  Meta-reasoning over multiple chains-of-thought

While meta-reasoning has proven effective on tasks that have qualitative evident information, its ability to stay consistent between arithmetic operations and its subsequent reasoning path witnesses the same limitations as baseline self-consistency and chain-of-thought [35].

### E.2  Importance Weighting with self-improvement

Unlike previously established self-improvement and Importance weighting methods as proposed by Jiang et al. [14]. Our methods weighs results directly after generation in a separate weighting/filtering step. While results showed some frailty if not tuned with fitting parameters we spare computational efforts by not requiring an addition pre-training step. Pre-trained self-improvement Models could be used together with our introduced weighting method, to test performance and facilitate accuracy even further

## F   Sample analysis

In the evaluation of AQuA-RAT, some results exhibited noise. Particularly smaller models failed to consistently follow the few-shot chain-of-thought examples occasionally. This led to instances where outputs could not be parsed for final analysis. To ensure reproducibility, we employed the parsing extraction approach from baseline self-consistency. Furthermore, some models showed degeneration after generating the initial response, highlighting the need of development for a custom extraction function to ensure accurate semantic interpretation, particularly when utilizing functions that include embeddings.

## G   Efficiency Comparison

Other than self-consistency our methods require additional computation, other than the initial generation to compute its results.

- **Embeddings**: The computational cost is moderate, as the BERT model utilized is of a manageable size, keeping resource usage at a reasonable level.

- **Centroid Proximity Weighting**: This method is computationally inexpensive, as it relies solely on mathematical operations without requiring extensive resources.

- **Semantic Consensus Weighting**: Similarly, this technique is computationally efficient due to its reliance on lightweight mathematical computations.

- **Outlier Detection**: All three outlier detection methods employed are computationally low-cost, ensuring minimal impact on overall performance.

Compared to baseline self-consistency, the performance loss is minimal on modern GPUs, with the most computational effort still lying on the initial generation. Additionally, unlike other self-improvement methods, our techniques don't require an extra pre-training phase and can be applied directly. As advancements in computational resources continue and smaller models grow increasingly capable, we expect this concern to become even less significant.

### G.1   Sampling from multiple temperatures

Baseline self-consistency samples of static temperature models often result in deterministic or overly random outputs. We sampled from five different temperatures per generation finding that it provides a wider range of outputs with a more diverse spectrum of answers and performs above average compared to baseline *self-consistency*.

| Method | Avg. Accuracy (%) |
|---|---|
| baseline SC | 46.50 |
| Varied temp. SC (MV) | 46.53 |
| Varied temp. SC (weight) | **48.54** |

Table 5: Weighted self-consistency with varying levels of abstraction improves performance over baseline.

It is to note that higher temperature showed a degree of randomness that can lead to higher degeneration. However this limiting factor can be mitigated when applied with inverse temperature weighting and improve performance of up to **2.5%**. The effect of different temperature sets can be found in Appendix N.

### G.2   Finetuned featurizers

The process of converting rationales into semantic embedding vectors was applied to multiple featurizer-models at different forms of fine-tuning to measure the ability of models to effectively convert sequences into fitting embedding vectors.

| BERT-Model | avg distance ($\downarrow$) | |
|---|---|---|
| RoBERTa | 48.697 | |
| MathBERT | 45.892 | **(-2.8)** |
| SciBERT | 45.281 | **(-3.4)** |

Table 6: Featurizers finetuned on similar distributions tend to pack answers more tightly together

The results revealed elevated results for SciBERT and MathBERT [29] when compared to RoBERTa. This is likely due to RoBERTa's general robust training where in contrast, both MathBERT and SciBERT exhibit stronger performance[3]. We conjecture that this is due to their training data being more representative of the reasoning tasks that we evaluate on here [30]. This observation suggests that improper or "unfitting" fine-tuning reduces overall data point density, resulting in a loss of information within the produced vectors, and consequently hindering subsequent marginalization techniques [22].

### G.3 Secondary semantic evaluation methods

The implementation of $k$-means clustering[4] showed that regardless of the fact that reasoning can be improved by detailed mappings, clustering didn't attribute to enhance the quality of the semantic evaluation. Additionally we reason this to be attributed to two limiting factors: We experimented with a spectrum of values for the parameter $k$, with a significant emphasis on $k$=2 to ensure that the clusters would still provide a sufficient amount of associated rationales with each cluster to utilize the effect of self-consistency.

Table 7: Performance using $k$-means for outlier detection, with $k = 2$

| Model | AQuA-rat | SVAMP | SQA |
|---|---|---|---|
| Llama 2 | 24.16 | 42.47 | 47.60 |
| Llama 3 | 46.06 | 72.33 | 17.6 |
| Mistral | 24.83 | 62.52 | 23.73 |
| GPT 3.5 | 65.52 | 78.67 | 21.97 |
| GPT-4o mini | 83.46 | 89.62 | 36.68 |

Table 8: Averaged over 10 runs, clustering has shown volatility based on initial cluster placement.

This method implies that the predictions associated with the majority cluster are the ones for which the model exhibits the greatest overall confidence. A detailed assessment of the found results can be accessed in Appendix M.1.

## H  N-Gram Rationale Comparison

### H.1  Rouge-N as a performance measure

Contrary to GPT 3.5's performance in terms of accuracy, it under performs in comparison when taking ROUGE metrics into account. As expected it excels in generating accurate, contextually relevant responses but expressed responses more detailed in a more comprehensive fashion, leading to lower ROUGE scores due to the strictly accurate less extensive rationale annotated in the dataset. [16]
The other Models like Llama 2 7B and Mistral 7B produce higher scores. This might be related to factors like style of writing and higher text length which although it leads to more comprehensive embeddings lowers it's score when compared with a metric like *Rouge-N* as visible in Table 3

---

[3]Tested on arithmetic samples only, due to their greater variability and problem-solving scope compared to the more logic-bound and less varied nature of coding tasks and QA tasks.

[4]Averaged over 10 random states to ensure an representative example.

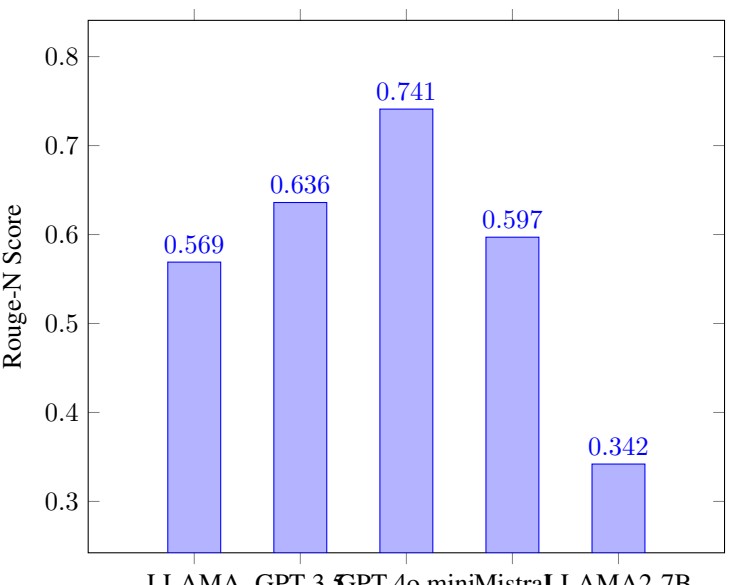

Figure 2: Average Rouge-N Scores across StrategyQA, AQuA-RAT, and SVAMP for Different Models

## H.2 N-Gram weighting

N-Grams are often used for context understanding, aiding tasks like sentiment analysis and language modeling In our study, we used N-Grams to weigh their impact on results, testing different 'n' values to see how they affect accuracy outcomes.

Table 9: Weighting results based on N-Gram overlap with n = 2

| Model | AQuA-RAT | SVAMP |
|---|---|---|
| Llama 2 | 15.5 | 32.8 |
| Mistral | 16.7 | 47.1 |
| GPT 3.5 | 25.3 | 63.9 |

The low accuracy and poor results, coupled with a degree of randomness in the result distribution, indicate challenges in effectively correlating text using N-Grams. We experimented with different values of 'n' for N-Grams, aiming to improve performance, but encountered limitations. As depicted in the table, the effectiveness of N-Grams varied, suggesting that the pure similar wording in rationales cant be utilized in an effective way to improve or even stably perform similar to the baseline. Higher values of "n" consecutively worsened results.

## I Configuration & Parameters

### I.1 Varying Response Count

Our analysis indicates that maintaining a minimum of 7-10 responses is crucial to achieving consistent performance comparable to the baseline. When $k$ is set to lower values, the performance gains diminish, sometimes leading to completely random results with low accuracy. We expect that increasing the number of responses could enhance the effectiveness of our methods, leveraging the additional context and range of responses available to each weighting mechanism for improved accuracy.

### I.2 Outlier detection Hyperparameters

#### I.2.1 k-nearest neighbor results

In the k-nearest neighbor (KNN) algorithm, parameters such as the number of neighbors (n_neighbors), the distance metric (metric), and the algorithm used for computing nearest neighbors were varied. The best-performing configuration in terms of accuracy was found with **n_neighbors set to 5**, using the **euclidean metric** using the **ball_tree algorithm** and a **threshold of 90%** that concluded to an averaged accuracy of **56.18%** with all Models and Datasets.

#### I.2.2 Isolation forest results

For the Isolation Forest, the grid search varied parameters including the number of estimators (n_estimators), the contamination factor, and the max samples size. The configuration yielding the highest accuracy utilized **n_estimators=200**, **contamination=auto**, and **max_samples=auto** with an performance of **58.56%** averaged across all Models and Datasets.

#### I.2.3 support vector machines results

In the case of Support Vector Machines (SVM), the kernel type (kernel), the regularization parameter (nu), and the gamma value were among the parameters adjusted. The most accurate results were achieved with a **linear kernel**, **nu set to 0.01**, and **gamma set to scale**. The average accuracy was **55.17%**

### I.3 Model configuration

- top-k: 50
- top-p: 50
- sampling: true
- max-new-tokens: see Appendix I.4
- temperature: see Appendix J.1

Configurations may deviate slightly on GPT 3.5 & GPT4o-mini due to usage via the public API.

### I.4 Token generation

We used a default of **250** max-new-tokens across all models on **SVAMP**, due to the complexity and length of sequences on **AQuA-RAT** we chose **400** max-new-tokens. **Humaneval** is known to cause degeneration after given stopwords, to limit potential faulty generation of new tokens to we set max new tokens to **400**. To ensure long enough reasoning chains we limited the generation on **StrategyQA** to **450** tokens.

## J Abstract consistency

### J.1 Temperature sets

We tested our theory of abstraction on a variety of temperature sets and found that *set 1* exhibits the best balance between diversity and correctness in our examples. Therefore, it outperforms the other proposed sets.

All other experiments have been conducted on a static *temperature* of **0.8** to aid reproducibility and comparability between results and effects of the employed mechanisms.

### J.2 Weighing abstract consistency

We propose several methods for weighing sequences from different temperatures. Additionally, we employ a weighing system based on the inverse of the applied temperature. Furthermore, we conducted tests using weighted squared inverse weighting on a small subset. However, these tests did not yield substantially elevated results and performed on a similar margin.

| Set 1 ($t$) | Set 2 ($t$) | Set 3 ($t$) |
|---|---|---|
| 0.9 | 0.7 | 0.5 |
| 0.8 | 0.6 | 0.4 |
| 0.7 | 0.5 | 0.3 |
| 0.6 | 0.4 | 0.2 |
| 0.5 | 0.3 | 0.1 |

Table 10: Each Temperature is tested on 1/5 of the samples per generation, to ensure an even distribution.

Figure 3: Average   Figure 4: Squared Average

$$\sum_{i=1}^{n} \frac{1}{t_i} \quad (1) \qquad \sum_{i=1}^{n} \left(\frac{1}{t_i}\right)^2 \quad (2)$$

# K   Prompting

Previous work in self-consistency indicated that chain-of-thought yielded the most favorable outcomes both in terms of accuracy and employed reasoning path. This strategy aligned well with the specific requirements and objectives of a long and sensical reasoning path. Therefore, we chose not to employ other prompting strategies as they did not match the required qualities as observed with chain-of-thought prompting.

## K.1   Used k-shot prompts

The used **8-Shot prompt** for mathematical reasoning follows the example provided in pg. 43 and use the on pg. 36 referenced set for AQuA on the AQuA-RAT dataset with of the original chain-of-thought paper.

---

Q: There are 15 trees in the grove. Grove workers will plant trees in the grove today. After they are done, there will be 21 trees. How many trees did the grove workers plant today?
A: We start with 15 trees. Later we have 21 trees. The difference must be the number of trees they planted. So, they must have planted 21 - 15 = 6 trees. The answer is 6.

Q: If there are 3 cars in the parking lot and 2 more cars arrive, how many cars are in the parking lot?
A: There are 3 cars in the parking lot already. 2 more arrive. Now there are 3 + 2 = 5 cars. The answer is 5.

Q: Leah had 32 chocolates and her sister had 42. If they ate 35, how many pieces do they have left in total?
A: Leah had 32 chocolates and Leah's sister had 42. That means there were originally 32 + 42 = 74 chocolates. 35 have been eaten. So in total they still have 74 - 35 = 39 chocolates. The answer is 39.

Q: Jason had 20 lollipops. He gave Denny some lollipops. Now Jason has 12 lollipops. How many lollipops did Jason give to Denny?
A: Jason had 20 lollipops. Since he only has 12 now, he must have given the rest to Denny. The number of lollipops he has given to Denny must have been 20 - 12 = 8 lollipops. The answer is 8.

Q: Shawn has five toys. For Christmas, he got two toys each from his mom and dad. How many toys does he have now?
A: He has 5 toys. He got 2 from mom, so after that he has 5 + 2 = 7 toys. Then he got 2 more from dad, so in total he has 7 + 2 = 9 toys. The answer is 9.

Q: There were nine computers in the server room. Five more computers were installed each day, from Monday to Thursday. How many computers are now in the server room?

A: There are 4 days from Monday to Thursday. 5 computers were added each day. That means in total 4 * 5 = 20 computers were added. There were 9 computers in the beginning, so now there are 9 + 20 = 29 computers. The answer is 29.

Q: Michael had 58 golf balls. On Tuesday, he lost 23 golf balls. On Wednesday, he lost 2 more. How many golf balls did he have at the end of Wednesday?
A: Michael initially had 58 balls. He lost 23 on Tuesday, so after that he has 58 - 23 = 35 balls. On Wednesday, he lost 2 more, so now he has 35 - 2 = 33 balls. The answer is 33.

Q: Olivia has $23. She bought five bagels for $3 each. How much money does she have left?
A: She bought 5 bagels for $3 each. This means she spent 5 * $3 = $15 on the bagels. She had $23 in the beginning, so now she has $23 - $15 = $8. The answer is 8.

---

**Proposed 4-shot on AQuA-RAT:**

---

Q: John found that the average of 15 numbers is 40. If 10 is added to each number then the mean of the numbers is?
Answer Choices: (a) 50 (b) 45 (c) 65 (d) 78 (e) 64
A: If 10 is added to each number, then the mean of the numbers also increases by 10. So the new mean would be 50. The answer is (a).

Q: If a / b = 3/4 and 8a + 5b = 22, then find the value of a.
Answer Choices: (a) 1/2 (b) 3/2 (c) 5/2 (d) 4/2 (e) 7/2
A: If a / b = 3/4, then b = 4a / 3. So 8a + 5(4a / 3) = 22. This simplifies to 8a + 20a / 3 = 22, which means 44a / 3 = 22. So a is equal to 3/2. The answer is (b).

Q: A person is traveling at 20 km/hr and reached his destination in 2.5 hr. Find the distance.
Answer Choices: (a) 53 km (b) 55 km (c) 52 km (d) 60 km (e) 50 km
A: The distance that the person traveled would have been 20 km/hr * 2.5 hrs = 50 km. The answer is (e).

Q: How many keystrokes are needed to type the numbers from 1 to 500?
Answer Choices: (a) 1156 (b) 1392 (c) 1480 (d) 1562 (e) 1788
A: There are 9 one-digit numbers from 1 to 9. There are 90 two-digit numbers from 10 to 99. There are 401 three-digit numbers from 100 to 500. 9 + 90(2) + 401(3) = 1392. The answer is (b).

Our generation on Humaneval was conducted **0-shot** using just the raw prompt given by the dataset.

# L   Datasets

We selected the datasets that are commonly used in similar methods such as baseline self-consistency [33] and related work to simplify reproduction and comparison to ensure consistency in our results.

We use the recommended configuration splits for testing as suggested by default for each dataset. For AQuA-RAT, our test set includes the full set of 254 examples. In the case of StrategyQA, we employ the complete test split, which consists of 687 samples. Specifically, for SVAMP, we utilize the train and test split comprising 1,000 samples to achieve a less noisy evaluation.

# M   K-means Clustering

Across our study we employed kmeans to cluster datapoints mapped by our featurizer model.

## M.1   Clustering effects

Clustering has shown diminishing returns in terms of accuracy, however the herein provided evidence shows that clustering with k-means provides a notable advantages which even tho the accuracy was low can be used as a diagnostic tool and similarity measure

### M.1.1 Silouhette score

We used the silhouette score to evaluate clustering effectiveness. This score measures how similar an object is to its own cluster compared to other clusters, ranging from -1 to 1.

Our obtained averaged silhouette score equals **0.41**, suggesting a moderate level of distinction between clusters. This range indicates that, on average, objects within a cluster are closer to each other than to objects in other clusters, but the separation is not highly distinct.

This finding suggests that clusters are indicating a clear structure in sentence and wording of results and due to Kmeans nature perform better on higher sample sizes.

### M.1.2 Average correct datapoint proportion

Despite the fragility shown during evaluation on benchmarks, the k-means accurately categorizes the majority of correct answer within the preponderant cluster, not only based on cluster size. This implies that the method, even with limited data, captures essential patterns effectively.

High-performing models are more likely to adhere closely to the chosen method. This is because when most answers are correct, there's a lower chance of incorrect responses outweighing the correct ones, which could lead to inaccuracies.

The shown results indicate a trend demonstrating that the selected cluster is more likely to feature the majority of correct responses, with an average of **60.5**%.

We witness the same strides towards higher sample sizes with the usage of k-means as already conveyed in the original self-consistency paper, here larger sample sizes might be able to capture the amount of correct answers in a more favorable manner due to their enabled potential for a higher number of clusters, capturing more nuanced and subtle variations rather than the broad range of responses.

### M.1.3 Cluster density comparison

The primary cluster and the ostensibly weaker, later-disregarded cluster exhibit comparable performance in terms of the average distance of the data points to its subsequent cluster centroid.

Table 11: Average Deviation for clusters

| Method | Model | Chosen cluster | Disregarded cluster |
|---|---|---|---|
| SVAMP | LLAMA 2 | 2.037 | 2.567 |
| SVAMP | Mistral | 2.981 | 3.800 |
| SVAMP | GPT 3.5 | 4.428 | 4.513 |
| SVAMP | GPT 4o mini | 4.356 | 4.653 |
| SVAMP | LLAMA 3 | 4.562 | 4.569 |
| AQuA-RAT | LLAMA 2 | 0.838 | 0.670 |
| AQuA-RAT | Mistral | 0.871 | 0.598 |
| AQuA-RAT | GPT 3.5 | 3.649 | 3.684 |
| AQuA-RAT | GPT 4o mini | 2.134 | 3.082 |
| AQuA-RAT | LLAMA 3 | 3.235 | 3.163 |
| StrategyQA | LLAMA 2 | 2.741 | 3.215 |
| StrategyQA | Mistral | 1.962 | 2.487 |
| StrategyQA | GPT 3.5 | 4.283 | 4.751 |
| StrategyQA | GPT 4o mini | 1.869 | 2.935 |
| StrategyQA | LLAMA 3 | 2.864 | 3.124 |

## N  Abstract consistency on different temperature sets

Higher temperature in generative models introduces a degree of randomness that can negatively impact performance by increasing degeneration in model outputs. However, this limiting factor can be partially mitigated through techniques such as inverse temperature weighting. When applied

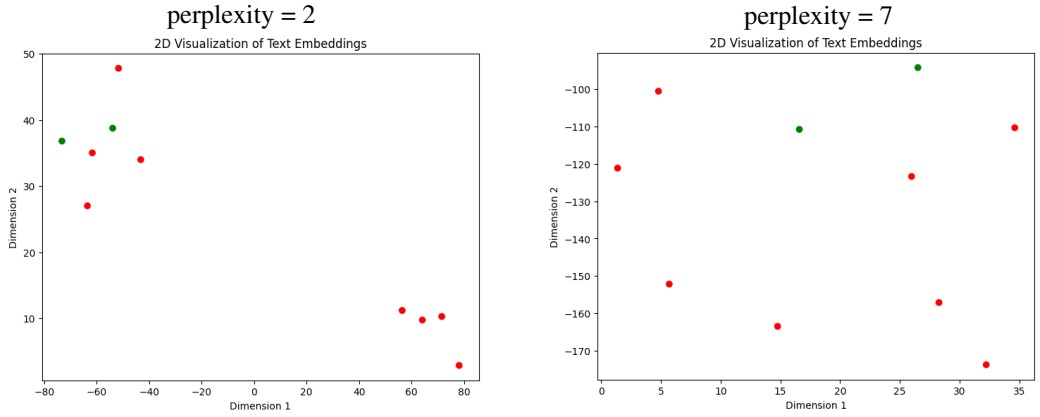

Figure 5: T-SNE reduced image based on a test on a subset of arithmetic reasoning examples, evaluated on 10, 15 and 20 generated outputs based on baseline self-consistency

appropriately alongside temperature variation. The benefits of higher temperature are not monotonic - beyond an optimal level, continuing to increase temperature will again degrade performance. There exists a sweet spot where judiciously elevated temperature and re-weighting allows models to produce greater diversity without excessive degradation which we found to lay between t = *0.5* and t = *0.9*.

# O   Dimensionality reduction

Dimensionality reduction did improve performance in edge cases, but it should not be relied upon for consistent results and was generally unstable. We recommend that our methods be used without any additional reduction to ensure more reliable and consistent outcomes. [26, 11, 15]

## O.1   t-SNE

To enhance separation and clustering in t-SNE for data exploration and pattern recognition tasks, we use a perplexity parameter of 2. This choice is based on the fact that local distributions in out scenario provide a more informative representation than global distributions due to the increased density of points in close proximity, which improves the detail captured in the mapping.

## O.2   PCA

In our scenario, while PCA might be better under very specific random circumstances, such as when linear relationships dominate the data, t-SNE is generally superior for visualization. t-SNE excels in revealing complex structures and patterns by capturing local relationships, making it more suitable for understanding the data visually.

