# OpenReview forum: "Semantic Self-Consistency: Enhancing Language Model Reasoning via Semantic Weighting"
_NeurIPS.cc/2024/Workshop/MATH-AI — MATH-AI 24_

### Official Review · Reviewer_w4a2 · 2024-10-06
**It is a strong paper that makes a valuable contribution to enhancing language model reasoning capabilities. The novel techniques are well-justified, the experimental evaluation is rigorous, and the analyses provide useful insights**

**Rating:** 7
**Confidence:** 4

**Review:**

Quality:

This paper presents a solid extension to the self-consistency framework for improving language model reasoning. The authors propose two main techniques - Centroid Proximity Weighting (CPW) and Semantic Consensus Weighting (SCW) - to incorporate semantic analysis of reasoning paths in addition to final decisions. The methodology is well-explained and the experimental setup covers multiple datasets and models.

The quality of the work is good, with a clear motivation, well-designed experiments, and thorough analysis of results. The authors provide extensive ablation studies and additional analyses in the appendix, demonstrating rigorous investigation.

Clarity:

The paper is clearly written overall and follows a logical structure. The methodology section explains the proposed techniques in sufficient detail. The results are presented clearly in tables and analyzed thoughtfully.

Some minor points could be clarified further:

More explanation of why certain datasets were chosen would be helpful.

The distinction between CPW and SCW could be explained more explicitly early on.

Some of the mathematical notation in Section 4 could benefit from more explanation

However, these are minor issues and overall the paper communicates the key ideas and findings effectively.

Originality:

The paper presents an original extension to the self-consistency framework by incorporating semantic analysis of reasoning paths. While building on existing work, the proposed techniques of CPW and SCW are novel contributions. The idea of using embedding vectors to analyze and weight reasoning paths is innovative in this context. The exploration of outlier detection methods for filtering responses is also an original angle.

Significance:

This work makes a meaningful contribution to improving language model reasoning capabilities. The proposed techniques show consistent improvements over baseline self-consistency across multiple datasets and models. The gains are particularly notable on more complex reasoning tasks like StrategyQA.
The paper provides useful insights into the effectiveness of semantic analysis for enhancing reasoning, which could inspire further work in this direction. The extensive ablation studies and analyses offer valuable findings for researchers working on language model reasoning.

Pros:

Novel techniques (CPW and SCW) for incorporating semantic analysis of reasoning paths
Consistent improvements over baseline self-consistency across multiple datasets/models
Thorough experimental evaluation and ablation studies
Insightful analysis of results and failure cases
Extensive additional studies and analyses provided in appendix

Cons:

Could provide more motivation/context for choice of datasets
Some explanations of techniques could be clearer/more detailed
Limited exploration of potential negative impacts or ethical considerations
Lacks comparison to some other recent techniques for improving LM reasoning

Overall, this is a strong paper that makes a valuable contribution to enhancing language model reasoning capabilities. The proposed techniques are novel and well-justified, the experimental evaluation is thorough, and the analyses provide useful insights. While there are some minor areas for improvement in clarity and broader impact discussion, the core technical contribution is solid and significant.

---

### Official Review · Reviewer_rggG · 2024-10-06

**Rating:** 5
**Confidence:** 4

**Review:**

Summary:
The paper introduces a new approach extending the self-consistency work of Wang et al by looking at the semantic consistency using multiple approaches between different rationales generated on the same input. The author(s) suggest utilizing outlier removal algorithms before taking a final majority vote on the model's response. The approach, while producing useful results in the specific limited context that the author(s) have tested in, lacks generalizability and could do with deeper investigations around the methodology itself as well as the candidate LLMs.

Strengths:
1. The author(s) propose a novel approach of analyzing the semantic relationships and similarities between responses generated by an LLM for the same prompts, combining Chain-of-thought reasoning with word embeddings.
2. The proposed approach of Semantic Consensus Weighting shows promising results for a subset of mathematical tasks and the results have been presented in an intuitive way.
3. The author(s) acknowledge the limitations of the proposed approaches.

Major concerns:
1. Details : Additional details on the size of the responses generated for each input would be helpful. It can also help with creating more diverse responses.
2. Robustness : Number of responses generated per input seems like an obvious hyper-parameter to try experimenting with to see if the performance of techniques improve.
3. Conclusion : Conclusion lacks a clear recommendation, even if the generalizability of the results is limited.
4. Clarity : No rationale/ explanation is provided on why some techniques work on some models and do not on others. This would be an important addition to the conclusion section and would help solidify use cases where authors' findings can be valuable.

Minor concerns:
1. Length of the paper exceeds 4 pages

Recommendation:
Author(s) must at least address major concerns before the paper can be accepted.

---

### Official Review · Reviewer_h4fq · 2024-10-06
**Promising Weighting Mechanism for Self-Consistency Framework but Need More Clarification**

**Rating:** 6
**Confidence:** 2

**Review:**

This paper introduces a novel approach to enhance reasoning in large language models through semantic similarity weighting and outlier detection. The contribution is promising, as it aims to improve reasoning coherence across multiple generated chains, but there are notable areas for improvement.

Advantages
- Innovative Weighting Method: The idea of weighting chains based on their semantic similarity is novel and potentially impactful for improving complex reasoning.
- Evaluation Across Multiple Datasets: The empirical results provide evidence of improved reasoning on datasets like AQuA-RAT and StrategyQA.
- Clear Methodology: The paper provides a step-by-step description of the proposed approach, making it easier for readers to understand the experimental setup, semantic weighting techniques, and evaluation metrics.

Areas for Improvement
- Inconsistent Performance Gains: While the proposed methods show improvement in some cases, results are inconsistent (e.g., CPW shows a drop on StrategyQA for GPT-4o mini). It would be better if the paper could address this by ablation studies or more explanation of these observations.
- Diversity vs. Coherence Trade-Off: The emphasis on semantic similarity may penalize diverse reasoning paths that reach the same conclusion from different angles. The paper would be more solid it it can explore this trade-off and its impact on final predictions.
- Lack of Interpretability: Since semantic embeddings may not perfectly capture reasoning, adding visualizations or case studies of how different reasoning paths are weighted would enhance transparency and help readers understand the method’s decision-making process.

---

### Official Review · Reviewer_u8CQ · 2024-10-07

**Rating:** 7
**Confidence:** 4

**Review:**

### Summary
This work proposes consistency and filtering mechanisms in the intermediates steps of reasoning steps of LLMs.


### Strengths

1. Strong indication of performance improvement.
2. The paper presents the motivation behind considering different weighting metrics and anomaly detection algorithms.

### Weaknesses

1. The paper misses some related developments in the weighting and filtering of LLM generation literature (Jiang, Chunyang, et al.).
2. The analysis only focuses on performance, and so the trade-offs (time or memory or API cost) are unclear when compared to baselines.


Jiang, Chunyang, et al. "Importance Weighting Can Help Large Language Models Self-Improve." arXiv preprint arXiv:2408.09849 (2024).

Minor:
Broken latex in Appendix D.

---

### Decision · Program_Chairs · 2024-10-08

**Decision:**

Accept

**Comment:**

The work presents an interesting technique to improve the reasoning abilities of LLMs and will be of interest to attendees of the workshop.